# Myeloblasts transition to megakaryoblastic immunophenotypes over time in some patients with myelodysplastic syndromes

**Kiyoyuki Ogata** *, Yuto Mochimaru, Kazuma Sei, Naoya Kawahara, Mika Ogata, Yumi Yamamoto

Department of Hematology, Metropolitan Research and Treatment Centre for Blood Disorders (MRTC Japan), Tokyo, Japan

* ogata@MRCJAPAN.com

## Abstract

### Objectives

In myelodysplastic syndromes (MDS), neoplastic myeloblast (CD34+CD13+CD33+ cells) numbers often increase over time, leading to secondary acute myeloid leukemia (AML). In recent studies, blasts in some MDS patients have been found to express a megakaryocyte-lineage molecule, CD41, and such patients show extremely poor prognosis. This is the first study to evaluate whether myeloblasts transition to CD41+ blasts over time and to investigate the detailed immunophenotypic features of CD41+ blasts in MDS.

### Methods

We performed a retrospective cohort study, in which time-dependent changes in blast immunophenotypes were analyzed using multidimensional flow cytometry (MDF) in 74 patients with MDS and AML (which progressed from MDS).

### Results

CD41+ blasts (at least 20% of CD34+ blasts expressing CD41) were detected in 12 patients. In five of these 12 patients, blasts were CD41+ from the first MDF analysis. In the other seven patients, myeloblasts (CD34+CD33+CD41- cells) transitioned to megakaryoblasts (CD34+CD41+ cells) over time, which was often accompanied by disease progression (including leukemic transformation). These CD41+ patients were more frequently observed among patients with monosomal and complex karyotypes. CD41+ blasts were negative for the erythroid antigen, CD235a, and positive for CD33 in all cases, but CD33 expression levels were lower in three cases when compared with CD34+CD41- blasts. Among the five CD41+ patients who underwent extensive immunophenotyping, CD41+ blasts all expressed CD61, but two cases had reduced CD42b expression, three had reduced/absent CD13 expression, and three also expressed CD7.

**Data Availability Statement:** All relevant data are within the paper and its Supporting Information files.

**Funding:** The authors received no specific funding for this work.

**Competing interests:** The authors have declared that no competing interests exist.

## Conclusions

Myeloblasts become megakaryoblastic over time in some MDS patients, and examining the megakaryocyte lineage (not only as a diagnostic work-up but also as follow-up) is needed to detect CD41+ MDS. The immunophenotypic features revealed in this study may have diagnostic relevance for CD41+ MDS patients.

## Introduction

Myelodysplastic syndromes (MDS) are heterogeneous myeloid malignancies with diverse prognoses ranging from indolent to rapidly progressive fatal courses [1]. The Revised International Prognostic Scoring System (IPSS-R), widely used for predicting prognosis in MDS, classifies patients into five categories based on the blast percentage in the bone marrow (BM), cytogenetic data, and depth of cytopenia [2].

In most cases, MDS blasts show a CD34+CD13+CD33+ myeloid progenitor immunophenotype [3–6]. Meanwhile, recent studies by us and others showed that blasts from a fraction of patients with MDS express CD41 (a marker of the megakaryocyte/platelet [MK/PLT] lineage) [7–9]. These patients (referred to as CD41+ MDS in this manuscript) often have monosomal and complex karyotypes and show poor prognosis, even without these poor prognostic karyotypes. These studies applied flow cytometry (FCM) using the CD45-blast gating method to detect CD41-positive blasts. As false positivity of CD41 in FCM can occur if platelets adhere to blasts, these studies applied various techniques to rule out this false positivity, including immunohistochemistry, cytoplasmic staining, examination of cytospin preparations of FCM samples, and electron microscopy.

In 1995, it was reported that GPIIb (CD41) mRNA was detected in patients with refractory anemia with excess blasts (RAEB) or RAEB in transformation (RAEB-t) but not in low-grade patients with MDS [10]. In acute megakaryoblastic leukemia (AMKL), blasts express MK/PLT lineage antigens and myeloid antigens (CD13/CD33) [11]. It has been reported that patients with AMKL often have complex karyotypes and, for more than half of the cases, are secondary to MDS or other antecedent hematologic disorders [12, 13]. Taken together, we speculate that myeloblasts shift to megakaryoblasts with disease progression in a fraction of patients with MDS, which eventually leads to a leukemic stage.

In this study, we serially analyzed the immunophenotypes of blasts from patients with MDS using multidimensional flow cytometry (MDF). This is the first report to reveal changes in immunophenotype of MDS blasts over time, from myeloblastic to megakaryoblastic, and the detailed immunophenoytypic features of CD41+ blasts.

## Materials and methods

### Patients

We studied patients with MDS and acute myeloid leukemia (AML; originating from MDS and had low blast counts [less than 30%]), who received azacytidine (AZA) therapy and their blasts were analyzed for CD41 expression via 6-color MDF at our institution from January 2019 to April 2022. Since MDF analysis was not repeated for patients not receiving AZA therapy, such patients were excluded from this study (12 MDS patients in the study period). The patients in this study did not overlap with patients in our previous studies [7, 8]. From April 2021, AZA therapy was combined with venetoclax in patients with AML who did not respond well to

AZA alone. Diagnoses were made according to the 2016 World Health Organization (WHO) criteria [14]. Karyotypes were analyzed using the standard G-banding technique and interpreted according to the International System for Cytogenetic Nomenclature criteria [15]. The monosomal karyotype was defined by the presence of at least two autosomal monosomies or one autosomal monosomy with at least one structural abnormality, as reported by Breems et al. [16]. The IPSS-R was applied to patients according to a previous report [2]. Clinical and laboratory data were collected from electronic database and analyzed on September, 2022.

Our institution is a referral center for treating patients with hematological malignancies, mainly MDS, but not a transplant center. Therefore, only patients who require (or are soon likely to require) disease-modifying therapies and choose to receive chemotherapy rather than stem cell transplantation (SCT) were referred to us. Patients who changed their minds and opted to undergo SCT were referred to transplant centers. This study was approved by the institutional review board of our institution (IRB186), and the procedures were in accordance with the Helsinki Declaration of 1975, as revised in 2008. Written informed consent was obtained from all patients. All data were fully anonymized before we accessed them.

## Multidimensional flow cytometry (MDF)

MDF analysis was performed at various time points: at a minimum, at the start of AZA therapy, when clinical improvement was not observed after at least three cycles of AZA therapy, and when disease progression was suspected, such as upon appearance or increase in circulating blasts and/or worsening cytopenia. Karyotyping was repeated at each time point in the MDF analysis.

Aspirated BM cells, which were anticoagulated with sodium heparin, were evaluated using MDF. For cases in which BM could not be aspirated, we used heparinized peripheral blood (PB) if it contained ≥2% blasts. In the present cohort, seven cases used PB samples and their actual blast percentages in PB were 8%–14%. Cell samples were processed as previously described in detail [8]. The six-color antibody combination used to examine CD41-positive blasts was CD235a (glycophorin A, GPA)-fluorescein isothiocyanate (FITC), CD33-phycoerythrin (PE), CD45-peridinin chlorophyll protein (PerCP), CD41a-PE-Cyanin 7 (PE-Cy7), CD34-allophycocyanin (APC), and CD16-APC-Cy7. We regarded a patient as positive for each antigen when at least 20% of the gated cells showed higher fluorescence than the negative control. From October 2021, the following three types of 6-color antibody combinations were applied to patients whose blasts were confirmed to be CD41-positive by the first panel. Tube 1 (GPA-FITC, CD33-PE, CD13-PerCP-Cy5.5, CD41a-PE-Cy7, CD34-APC, and CD45-APC-H7), tube 2 (CD71-FITC, CD42b-PE, CD61-PerCP-Cy5.5 CD41a-PE-Cy7, CD34-APC, and CD45-APC-H7), and tube 3 (CD19-FITC, CD56-PE, CD7-PerCP-Cy5.5, CD41a-PE-Cy7, CD34-APC, and CD45-APC-H7). Prior antibody titrations were performed for each antibody. This is an important step for avoiding false-positive results for CD41. Detailed antibody information is shown in S1 Table. The stained samples were acquired using a FACSLyric flow cytometer and analyzed using FACSuite software (BD Biosciences, San Jose, CA, USA). The flow cytometer was calibrated using fluorescent beads (BD FC beads; BD Biosciences). Daily instrument quality controls were performed to ensure identical operation from day to day.

If platelets adhere to cells, it may result in a false-positive result for CD41. Also, as a rare event, if cell samples contain micromegakaryocytes and/or megakaryocyte fragments, they may be mistaken as CD41-positive blasts [17]. Therefore, we examined MDF data for other cell fractions, such as lymphocytes and CD34-negative myeloid cells, in addition to CD34 + blasts, to evaluate if CD41-positivity was observed in every cell fraction (evidence of

CD41-false positivity due to platelet adherence to cells). In addition, stained cells were used first for MDF, and residual cells were subjected to cytospin preparations to examine whether platelets adhered to blasts and other cells and whether micromegakaryocytes and/or megakaryocyte fragments were present in the samples. In this series of MDF analyses, we did not observe any samples with micromegakaryocytes and/or megakaryocyte fragments. We found three samples that showed CD41-positivity on lymphocytes and CD34-negative myeloid cells, in addition to blasts, by MDF, and their cytospin slides confirmed platelet adhesion to blasts and other cells. We excluded these three cases from the present data.

### Imaging flow cytometry

An Attune CytPix Flow Cytometer (Thermo Fisher Scientific Co., Fair Lawn, NJ, USA) was used to analyze the selected samples. This is a machine of imaging flow cytometry (IFC), which captures digital images of cells at single-cell resolution and standard FCM data simultaneously. Thus, it is capable of examining cell images of every cell fraction analyzed by MDF, such as CD41-positive and -negative cells in every type of cell fraction. The cell samples for IFC were stained with CD33-PE, CD41a-PE-Cy7, CD34-APC, and CD45-APC-H7. Data were analyzed by Attun Cytometric Software (Thermo Fisher Scientific Co.) and FlowJo software (BD Biosciences).

### TP53 mutation analysis

Genomic DNA was extracted from aspirated BM cells or PB in the seven cases described in the MDF section. The DNA were analyzed for TP53 mutations by direct sequence analysis as previously described at the laboratory of the Faculty of Medicine, Shimane University [18, 19]. Briefly, DNA was amplified by polymerase chain reaction using primers for exons 4–9. The sequence was determined based on the dideoxy terminator method using an ABI PRISM 3130xl Genetic Analyzer (Applied Biosystems, Foster City, CA, USA) according to the manufacturer's protocol. TP53 mutation analysis was performed at the initial MDF analysis for all patients, and repeated for patients in which blasts became CD41-positive with time.

### Statistical analyses

Differences in categorical variables and continuous variables were evaluated using two-sided Fisher's exact tests and Mann-Whitney U test, respectively, using GraphPad Prism 8.3 (GraphPad, La Jolla, CA, USA).

## Results

### Identification of CD41+ patients with MDS

Fig 1 shows a representative example of identifying CD41+ patients with MDS (Case 1). Most CD34+ cells expressed CD33 and 26% of CD34+ cells expressed CD41 (Fig 1C). Other cell fractions (two CD34 dull or negative myeloid cell fractions [Fig 1E and 1F] and lymphocytes [G]) did not express CD41. The CD34+CD41+ cells were negative for GPA (Fig 1D) and CD16. Cytospin slides prepared using residual cells from the MDF tube, which were stained with 6-colors and used for data acquisition, showed that platelets did not adhere to cells, and micromegakaryocytes and megakaryocyte fragments were absent (Fig 1H). Fig 2 shows another example (Case 2). As in Case 1, CD34+ cells, but not other cell fractions, expressed CD41 (CD41+ cells, 37% of CD34+ cells), and cytospin slides showed neither platelet adherence to cells nor the presence of micromegakaryocytes and megakaryocyte fragments. The difference between these two cases was that CD34+CD41+ cells showed lower CD33 expression

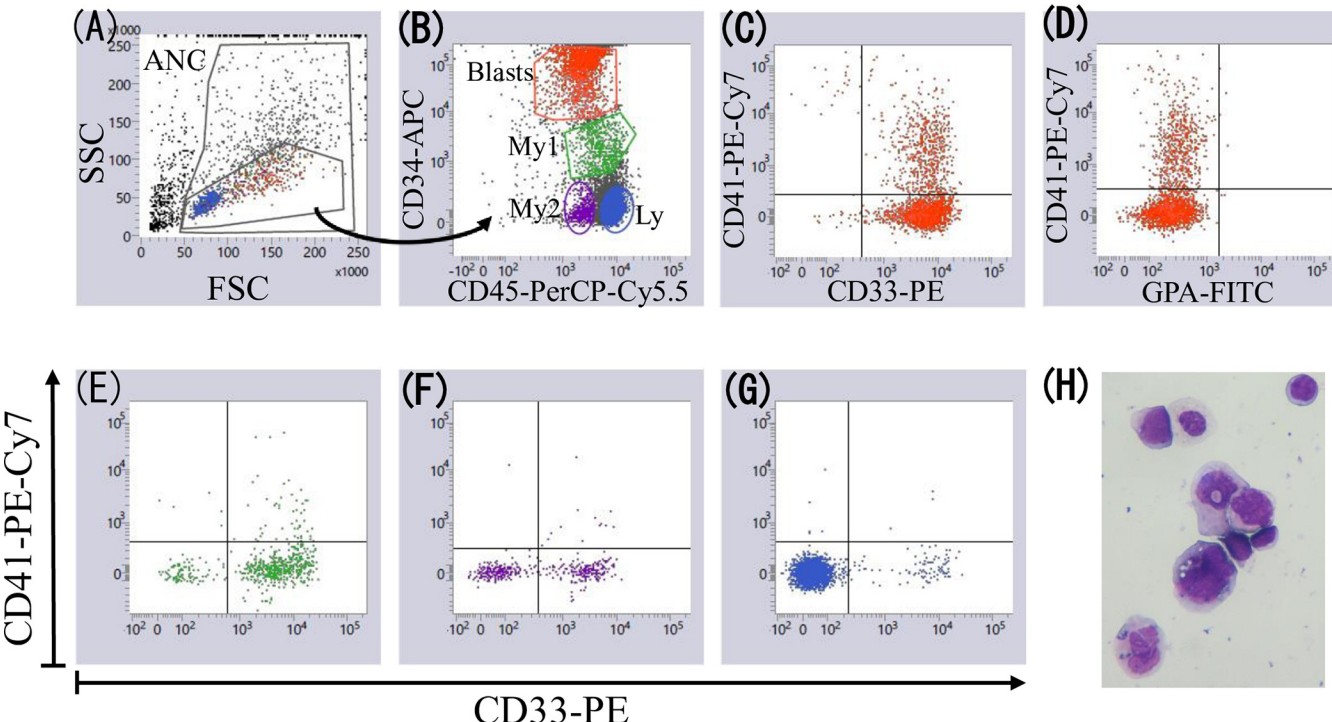

**Fig 1. Identification of CD41+ MDS (Case 1). A:** Singlet events were displayed on a forward scatter (FSC) versus side scatter (SSC) plot, and cells with a low SSC were gated. ANC comprises all nucleated cells. **B:** The gated low-SSC cells in panel A were displayed on the CD34 versus CD45 plot. CD34+ blasts (red dots) and three other cell fractions (two CD34 null or negative myeloid cell fractions [green and purple dots] and lymphocytes [blue dots]) were gated. **C and D:** CD34+ blasts were positive for CD33 and CD41 and negative for GPA. **E-G:** The other three cell fractions were negative for CD41. **H:** Cytospin slides show that platelets did not adhere to cells, and micromegakaryocytes and megakaryocyte fragments did not exist.

than CD34+CD41− cells in Case 2. The MDF data plots of these cases are shown in more detail in S1 Fig.

## Characteristics of all patients and CD41+ patients

Seventy-four patients met criteria for inclusion. The patient characteristics of these patients in the first MDF analysis are shown in Table 1. Since our institution is a referral center for treating advanced stages of MDS, most subjects were EB-1 or more advanced, and 67 of 74 patients were high or very high in IPSS-R. The *TP53* gene was mutated in 11 patients, and nine of these 11 patients had complex karyotypes defined by three or more karyotype abnormalities.

In this cohort, CD41+ blasts (at least 20% of CD34+ cells expressing CD41) were detected in 12 patients; five patients were CD41-positive from the first MDF analysis, and the other seven patients became CD41-positive with time, which was detected in the serial analyses. The characteristics of these 12 patients, when CD41-positivity was documented, are summarized in Table 2, and a comparison of their basic characteristics with those of CD41-negative patients is presented in S2 Table. Consistent with previous studies, CD41+ patients with MDS were preferably observed in patients with monosomal and complex karyotypes compared with other patients (monosomal vs. others, P < 0.001) (complex vs. others, P = 0.005). Although both CD41+ blasts and TP53 mutations are often observed in patients with complex karyotypes, the overlap between these two risk factors is infrequent; only three CD41+ patients had TP53 mutations.

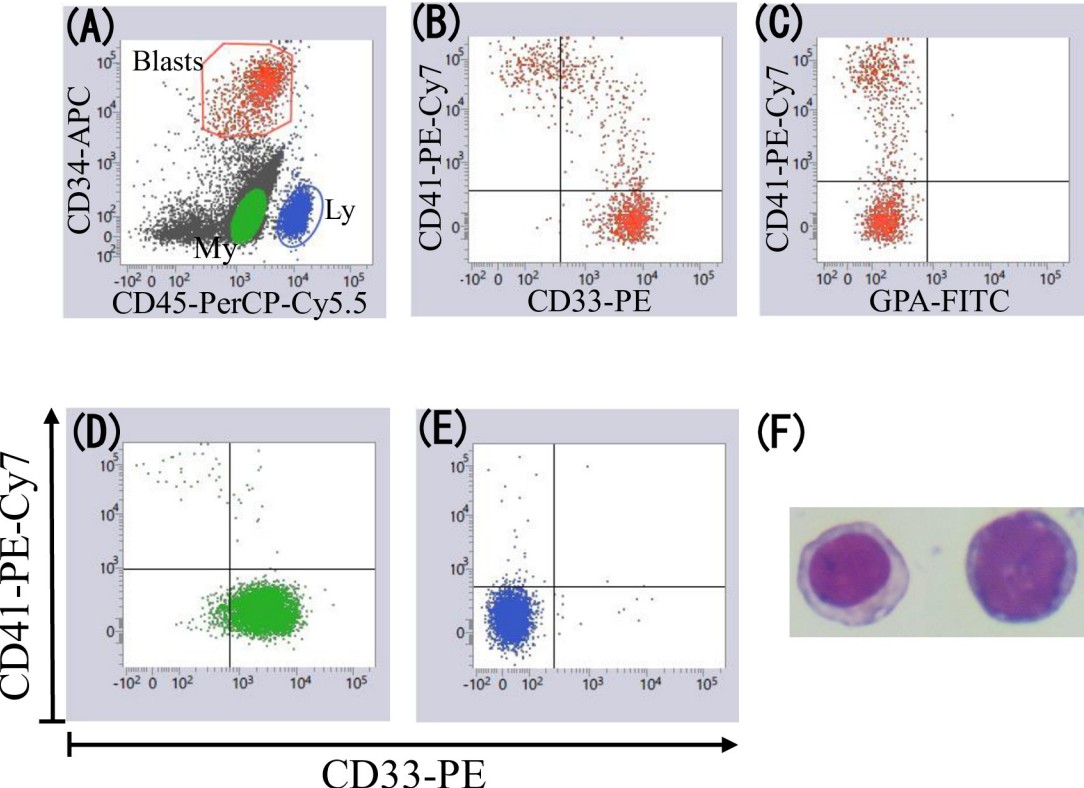

**Fig 2. Identification of CD41+ MDS (Case 2). A:** Cells with low SSC gated from singlet events were displayed on CD34 versus CD45 plots. CD34+ blasts (red dots) and other two cell fractions (CD34 negative myeloid cells [green dots] and lymphocytes [blue dots]) were gated. **B:** CD34+ blasts were positive for CD41. CD33 expression was lower in CD34+CD41+ cells than in CD34+CD41− cells. **C:** CD34+ blasts were negative for GPA. **D-E:** Other cell fractions were negative for CD41. **F:** Cytospin slides showing that platelets did not adhere to cells, and micromegakaryocytes and megakaryocyte fragments did not exist.

Regarding the expression of other antigens on CD41+ blasts in these 12 patients, all were negative for GPA and CD16 and positive for CD33. However, CD33 expression levels on CD34+CD41+ blasts were reduced compared with CD34+CD41- blasts in three patients, as judged by the pattern of CD33 versus CD41 plot shown in Fig 2. When the reduction in CD33 expression was expressed quantitatively by the ratio of the geomean values of CD33 expression (data from CD34+CD41+ cells divided by data from CD34+CD41- cells), this ratio was 0.106 for Case 2 and 0.162 and 0.358 for the other two cases.

We were able to analyze the immunophenotypes in more detail in five patients. CD41 + blasts expressed CD42b and CD61 in all five patients. However, when expression levels of CD42b, a maturation-associated molecule in megakaryopoiesis [20], were compared between CD41+ blasts and platelets in the same patients, it was reduced in CD41+ blasts in two patients (the MDF pattern of one of these two cases is shown later). The expression levels of CD41 and CD61 show a strong correlation in all five patients examined (S2 Fig); this correlation between CD41 and CD61 is also observed in normal platelets, since CD41 and CD61 molecules form a gpIIb/IIIa complex. Regarding CD13, CD41+ blasts showed reduced or absent CD13 expression in three cases. The association between CD13 and CD33 expression in CD34+CD41 + blasts in these three cases is as follows: CD13 and CD33 expression was reduced compared to CD34+CD41- blasts in one case, CD13 expression was absent in all CD34+ cells that were positive for CD33 in one case, and CD13 expression was absent in CD34+CD41+ blasts but

**Table 1. Characteristics of subjects at the first MDF analysis.**

| Variables | | Value |
|---|---|---|
| Male/Female | | 60/14 |
| Age, years | | 36-84 (median 69) |
| Therapy-related (yes/no) | | 10/64 |
| Diagnosis | | |
| | MLD | 2 |
| | EB-1 | 17 |
| | EB-2 | 27 |
| | AML | 28 |
| IPSS-R | | |
| | Low | 2 |
| | Intermediate | 4 |
| | High | 18 |
| | Very high | 49 |
| Cytogenetic category* | | |
| | Very good | 2 |
| | Good | 27 |
| | Intermediate | 21 |
| | Poor | 7 (1) |
| | Very poor | 16 (10) |
| Monosomal karyotype | | |
| | Yes | 11 |
| | No | 62 |
| TP53 gene | | |
| | Mutated | 11 |
| | Normal | 63 |

MDF, multidimensional flow cytometry; MLD, multilineage dysplasia; EB, excess blasts; AML, acute myeloid leukemia; IPSS-R, revised international prognostic scoring system.

The data in parentheses are number of patients with monosomal karyotype.

*Defined by IPSS-R.

The IPSS-R and cytogenetic data were missing in one case.

not in CD34+CD41- blasts in one case in which CD34+CD41+ blasts were positive for CD33. CD7, a marker reported to be associated with poor prognosis in MDS and AML [3, 21], was expressed in CD34+CD41+ blasts in three cases.

## Emergence of CD41+ blasts in the sequential analyses

The data of seven patients who became CD41-positive over time in the serial analyses are summarized in Table 3. Four to 30 months after the first MDF analyses in which blasts were negative for CD41, blasts became CD41-positive, often accompanied by disease progression. Two EB-2 cases (Cases 2 and 3) developed AML, two MLD cases (Cases 4 and 5) developed EB-1, and one remission case (Case 7) relapsed when CD41-positivity was documented. The cytogenetic category worsened in two cases (Cases 2 and 5) at the time of the CD41+ phase. Five of the seven patients died 1–14 months after CD41+ documentation. Case 6 survived 9 months after the CD41 documentation, but his karyotype category became poor in the last karyotyping. Examples of the emergence of CD41+ blasts in sequential MDF data are shown in S3 Fig.

Table 2. Characteristics of CD41+ patients.

| Variable | | | Value |
|---|---|---|---|
| Male/Female | | | 9/3 |
| Age, yrs (range and median) | | | 52-80 (64) |
| WHO | | | |
| | EB-1 | | 4 |
| | EB-2 | | 4 |
| | AML | | 4 |
| IPSS-R | | | |
| | High | | 3 |
| | Very high | | 9 |
| Karytoytypes | | | |
| | Normal | | 1 |
| | Monosomy 7 | | 1 |
| | Other single abnormalities | | 2 |
| | Three abnormalities | | 1 (1)* |
| | Four or more abnormalities | | 7 (6)* |
| TP53 mutation | | | |
| | Positive | | 3 |
| | Negative | | 9 |
| MDF data | | | |
| | CD41+ cells (%, range and median)** | | 26-55 (34) |
| | Other antigens*** | | |
| | | CD33 | 12 (3)/12 |
| | | GPA | 0/12 |
| | | CD16 | 0/12 |
| | | CD42b | 5 (2)/5 |
| | | CD61 | 5/5 |
| | | CD7 | 3/5 |
| | | CD13 | 3 (1)/5 |
| | | CD19 | 0/5 |
| | | CD56 | 2/5 |
| | | CD71 | 4/5 |

WHO, World Health Organization; EB, excess blasts; AML, acute myeloid leukemia; IPSS-R, revised international prognostic scoring system; MDF, multidimensional flow cytometry; GPA, glycophorin A.

All data were when CD41-positivity was documented.

*Data in parentheses in karyotypes are number of patients with monosomal karyotype.

**Positive cells in CD34+ blasts.

***Positive cases/analyzed cases. Positive was defined as 20% or more CD41+ cells expressed each antigen. Data in parentheses represent the number of patients with antigen expression lower than that in their own CD34+CD41-blasts (CD33 and CD13) and platelets (CD42b).

In Case 6, both blasts and monocytes were positive for CD41. Moreover, both CD41+ blasts and CD41+ monocytes in this case showed reduced CD42b expression compared with platelets in the same MDF tube (Fig 3). Cytospin slides showed neither platelet adherence to cells nor the presence of micromegakaryocytes and megakaryocyte fragments. These data indicate that both blasts and monocytes are truly positive for CD41 expression. To consolidate this finding further, we applied imaging FCM to analyze this sample.

**Table 3. Emergence of CD41+ blasts with time.**

| Case No. | Age/Sex | CD41-negative stage | | | Time from CD41- (Months) | CD41-positive stage | | | Last follow-up Time from CD41+ (Months) | Status |
|---|---|---|---|---|---|---|---|---|---|---|
| | | WHO | Cytogenetics Category | Monosomal | | WHO | Cytogenetics Category | Monosomal | | |
| 1 | 64/M | EB-2 | Very poor | Yes | 4 | EB-2 | Very poor | Yes | 3 | Dead |
| 2 | 54/M | EB-2 | Good | No | 28 | AML | Very poor | Yes | 1 | Dead |
| 3 | 52/M | EB-2 | Poor | Yes | 21 | AML | Poor | Yes | 5 | Dead |
| 4 | 58/M | MLD | Intermediate | No | 30 | EB-1 | Intermediate | No | 8 | Dead |
| 5 | 56/M | MLD | Intermediate | No | 14 | EB-1 | Very poor | No | 14 | Dead |
| 6 | 64/M | EB-1 | Intermediate | No | 7 | EB-1 | Intermediate* | No | 9 | Alive |
| 7 | 64/F | EB-2 in CR | Good | No | 13 | EB-2 | Good | No | 18 | Alive |

WHO, World Health Organization; EB, excess blasts; AML, acute myeloid leukemia; MLD, multilineage dysplasia; CR, complete remission.

*Cytogenetic category of this case changed to poor at the last follow-up time.

First, we analyzed a sample from a different patient, which showed false positivity for CD41, as a control for detecting CD41-false positivity in imaging FCM (S4 Fig). In this control sample, both blasts and lymphocytes showed CD41-positive in FCM, and cytospin slides showed platelet adherence to various cells. FCM imaging showed that platelets often adhere to CD41+ lymphocytes and CD41+ blasts, but not to CD41- lymphocytes and CD41- blasts. Since an imaging FCM captures cell images from one direction, platelets adhering to the opposite surface of cells were not detected. However, platelet adherence to cells was observed in CD41+ (false-positive) cell fractions alone and never in CD41- cell fractions in this control sample.

When the sample from Case 6 was analyzed by FCM, platelet adherence was not detected in either the CD41+ or CD41- fractions from monocytes and blasts (S5 Fig).

## Discussion

In general, MDS blasts show a CD13+CD33+CD34+ phenotype with a relatively low frequency of aberrant antigen expression, such as CD7, CD11b, and CD56 [3–5, 22]. Recent data from two separate groups, including ours, showed that CD41+ blasts were detected in a fraction of patients with MDS, and these patients showed poor prognosis and were often associated with the presence of monosomal and complex karyotypes [7–9]. This study is the first to demonstrate that immunophenotypes of MDS blasts acquire MK/PLT lineage markers over time in a fraction of the patients. Also, we revealed immunophenotypes of CD41+ blasts more detailed than before.

Previous studies have shown that patients with AMKL often have a history of MDS and complex karyotypes [12, 13]. The reported immunophenotype data from AMKL blasts show that one or more of the platelet glycoproteins (CD41, CD42b, and CD61) are positive, CD13 and CD33 may be positive, CD34 is often positive in adults but negative in children, and CD7 may be positive [11, 23]. These data were consistent with the present data for CD41+ MDS blasts. Taken together, it is reasonable to assume that in a fraction of patients with MDS, myeloblasts acquire megakaryoblastic immunophenotypes over time, eventually leading to secondary AMKL. We identified a case in which monocytes expressed MK/PLT antigens. However, because monocytes are often clonal in MDS, this finding is not surprising.

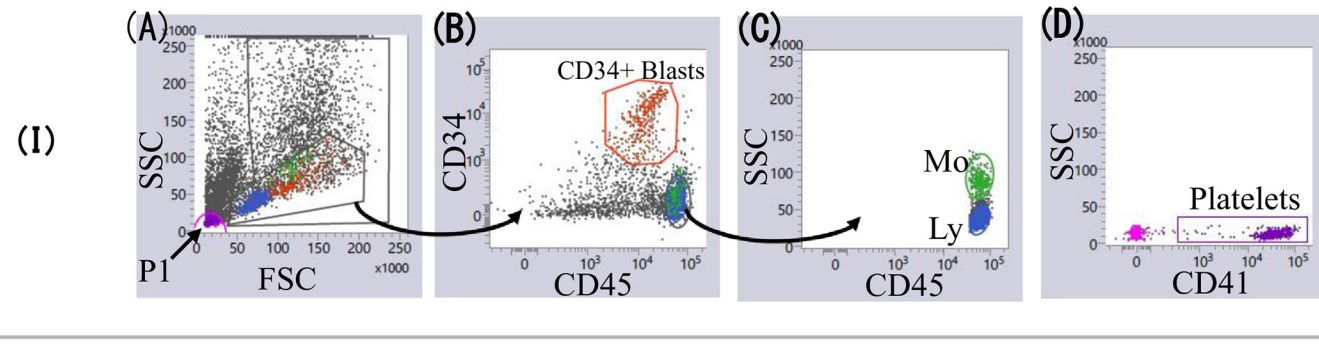

**Fig 3. MDF data from Case 6. [Lane I]. A:** Singlet events were displayed on the FSC versus SSC plot, and cells with low SSC were gated. A P1 gate was used to identify platelets. **B:** Low SSC cells (gated in panel A) were displayed on the CD34 versus CD45 plot. CD34+ blasts (red dots) and CD45-bright cells were gated. **C:** CD45-bright cells in Panel B were displayed on SSC versus CD45 plots, and lymphocytes (blue dots) and monocytes (green dots) were gated. **D:** Events in P1 (Panel A) were displayed on SSC versus CD41 plots and platelets were gated (purple dots). **[Lane II].** CD34+ blasts (A and C) and monocytes (B and D) were positive for CD13, CD33, and CD41. **[Lane III].** CD34+ blasts (A) and monocytes (B) were positive for both CD41 and CD42b, and their CD42b expression levels were lower than those in platelets (C). Lymphocytes were negative for CD41 and CD42b (D). **[Lane IV].** CD34+ blasts (A), monocytes (B), and platelets (C) were positive for both CD41 and CD61, corresponding to the fact that CD41 and CD61 molecules form a gpIIb/IIIa complex. Lymphocytes were negative for CD41 and CD61 (D).

Our data indicate that the acquisition of megakaryoblastic immunophenotypes in MDS is not very rare, with 12 of the 67 patients with high/very high risk in IPSS-R in our cohort. To diagnose CD41+ MDS, false-positive results for MK/PLT antigens should be carefully eliminated. For this purpose, examining MDF data from other cell fractions, such as lymphocytes and mature myeloid cells, in addition to blasts, and combined analysis of CD42b with CD41 or CD61 have been described to be useful in diagnosing AMKL [20, 23]. Reduced or absent CD42b expression on blasts has been described in a fraction of patients with AMKL and is ascribed to the fact that CD42b is a maturation-associated molecule in megakaryopoiesis, and therefore still scanty on megakaryoblasts. The expression levels of CD41 and CD61 showed a strong correlation in all five patients examined in this study. However, more patients need to be examined to verify whether this strong correlation is universal in CD41+ patients with MDS. In AMKL, immunohistochemistry is often useful for confirming the diagnosis. In MDS, however, since the percentage of blasts is lower than that in AML, the interpretation of immunohistochemistry results is not straightforward, particularly when applying a 20% threshold as a positivity of MK/PLT antigens. In this regard, morphological assessment using cytospin slides and imaging FCM is useful for confirming the diagnosis.

This study confirmed the data from previous reports that CD41+ MDS was preferably observed in patients with monosomal and complex karyotypes [7–9]. Similar to CD41+ MDS, TP53 mutated patients with MDS often have monosomal and complex karyotypes and poor prognosis [24–26]. However, the co-occurrence of CD41+ blasts and TP53 gene mutations was not so often in this study. This finding is consistent with reported data that, among patients with complex karyotypes that are linked to poor prognosis, the presence of TP53 mutation was associated with poorer prognosis [26], while the presence of CD41+ blasts was not [7]. The frequency of TP53 gene mutations in adult AMKL has not been well studied. One study reported that six of 24 adult AMKL patients (25%) had TP53 mutations [27].

We have no clear explanation as to why CD41+ blasts emerge over time in patients with MDS. This emergence is often accompanied by disease and cytogenetic progression. Therefore, an MDS clone may transform into a more aggressive phenotype with the emergence of CD41+ blasts. In the traditional hematopoietic pathway, megakaryocytes are produced from hematopoietic stem cells (HSCs) through a hierarchical series of progenitor cells, including multipotent, common myeloid, and megakaryocyte-erythrocyte progenitors. Recent data suggests that megakaryocytes can be produced directly from HSCs, skipping the stepwise commitment process [28, 29]. Moreover, CD41-expressing HSCs have been observed, which are myeloid-biased and increase with age [30, 31]. Myeloid-biased HSCs may be a transformation stage in MDS and AML [32]. One hypothetical scenario is that transformed myeloid-biased HSCs, expressing CD41 but no other lineage markers, produce several subclones, one of which is dominant at a relatively early phase of MDS, and later, a high-grade clone, which is phenotypically close to the transformed HSC, becomes dominant. The fact that CD41 expression was accompanied by a reduction in CD13/CD33 expression in some patients may support this hypothesis.

There are points that must be clarified in relation to CD41+ MDS in future studies. Since all patients in this study were treated with AZA, it remains unknown if CD41+ blasts emerge over time in untreated patients. Second, because the present patient cohort consisted of vey high-risk Japanese patients, incidence and other characteristics should be examined in other cohorts. Moreover, in this study, serial bone marrow aspirates (immunophenotyping) were often omitted for patients showing a good response to AZA therapy. Therefore, this study did not reveal whether CD41 expression might also be found in patients at low risk or in those responding well to AZA. Meanwhile, previous studies have shown that CD41+ patients with MDS are often therapy-resistant and show poor prognosis, and that CD41+ is often associated

with complex and monosomal karyotypes [7–9]; the results of this study are in agreement with these findings. Therefore, the development of therapies targeting this group of patients is of great interest and importance. To achieve this, the accumulation of data from investigating these patients from different angles involving both basic and clinical research, is required. Here, we emphasize the importance of analyzing the MK/PLT lineage not only as a diagnostic work-up, but also as a follow-up study in patients with MDS.

## Supporting information

**S1 Fig. MDF data of Cases 1 and 2. Lane A:** Cells with low SSC were separated on the CD34 vs. CD45 plot, and CD34+ blasts (red dots), CD34-dull or -negative myeloid cells, and lympho-cytes were gated. **Lane B:** Data from cells stained with 6-color. CD34+ blasts (red dots) are positive for CD41 expression. **Lane C (negative control 1):** Data from cells stained with CD33-PE, CD45-PerCP, and CD34-APC. **Lane D (negative control 2):** Data from cells stained with CD45-PerCP and CD34-APC. **Lane E:** Data from cells stained with 6-color. The CD34 +CD41+ cells were negative for both GPA and CD16.
(DOCX)

**S2 Fig. The expression levels of CD41 and CD61 show a strong correlation in all five patients examined.** The data from four patients are presented. The data of one remaining case are shown in Fig 3.
(DOCX)

**S3 Fig. Emergence of CD41+ blasts with time in sequential analyses.** The data from three cases (Cases 2, 3, and 5 in Table 3) are shown. Panels A and B in each case show data when CD34+ blasts (red dots) were negative for CD41. The panels C-F in each case are data when CD34+ blasts (red dots) became positive for CD41 with time (D). Other cell fractions (green and blue dots) were negative for CD41 (E and F).
(DOCX)

**S4 Fig. A CD41 false-positive case analyzed by imaging FCM. [Left panel]** A cytospin slide prepared from the FCM sample showing that platelets adhered to various cells (arrows). **[Mid-dle panel] A:** Singlet events were displayed on the FSC versus SSC plot, and cells with low SSC were gated. A P1 gate was used to identify platelets. **B:** Low SSC cells (gated in panel A) were displayed on the CD34 versus CD45 plot. CD34+ blasts (red dots) and CD45-bright cells were gated. **C:** CD45-bright cells in Panel B were displayed on SSC versus CD45 plots, and lympho-cytes (blue dots) were gated. **D:** Events in P1 (Panel A) were displayed on SSC versus CD41 plots and platelets (CD41+ cells) were gated. **E:** CD34+ blasts were displayed on the CD41 versus CD33 plot. CD41+ blasts (BL41+) and CD41- blasts (BL41-) were gated. **F:** Lymphocytes were displayed on the CD41 versus CD33 plot. CD41+ lymphocytes (Ly41+) and CD41- lymphocytes (Ly41-) were gated. **[Right panel] Images of various cell fractions gated in the middle panel. A:** Platelets. **B:** CD41+ blasts. **C:** CD41- blasts. **D:** CD41+ lymphocytes. **E:** CD41- lymphocytes. The arrow head indicate RBC ghost, which is observed in every cell fraction. The arrows indicate platelets adhering to leukocytes, which caused false CD41-positivity in leukocytes. Platelet adher-ence was observed in CD41-positive cell fractions (B and D), but not in CD41-negative cell frac-tions (C and E). Note that because the imaging FCM captures cell images from one direction, platelet adherence to the back surface of leukocytes was not detected.
(DOCX)

**S5 Fig. Case 6 analyzed by imaging FCM. [Left panel] A:** Singlet events were displayed on the FSC versus SSC plot, and cells with low SSC were gated. A P1 gate was used to identify

platelets. **B:** Low SSC cells (gated in panel A) were displayed on the CD34 versus CD45 plot. CD34+ blasts (red dots) and CD45-bright cells were gated. **C:** CD45-bright cells in Panel B were displayed on SSC versus CD45 plots, and monocytes (green dots) were gated. **D:** Events in P1 (Panel A) were displayed on SSC versus CD41 plots and platelets (CD41+ cells) were gated. **E:** CD34+ blasts were displayed on the CD41 versus CD33 plot. CD41+ blasts (BL41+) and CD41- blasts (BL41-) were gated. **F:** Monocytes were displayed on the CD41 versus CD33 plot. CD41+ monocytes (Mo41+) and CD41- monocytes (Mo41-) were gated. **[Right panel] Images of various cell fractions gated in the left panel. A:** Platelets. **B:** CD41+ blasts. **C:** CD41- blasts. **D:** CD41+ monocytes. **E:** CD41- monocytes. Platelet adhesion was not observed in CD41-positive cell fractions (B and D), but also in CD41-negative cell fractions (C and E). (DOCX)

**S1 Table. List of antibodies used in this study.**
(XLSX)

**S2 Table. Comparison of basic characteristics between CD41+ and CD41- patients.**
(XLSX)

## Acknowledgments

We appreciate Takeshi Murooka, Saki Miyagawa, and Takashi Sekiguchi (Thermo Fisher Scientific Co.) for their valuable assistance in operating an imaging flow cytometer.

## Author Contributions

**Conceptualization:** Kiyoyuki Ogata.

**Data curation:** Kazuma Sei, Naoya Kawahara, Mika Ogata.

**Formal analysis:** Yuto Mochimaru.

**Investigation:** Kiyoyuki Ogata.

**Methodology:** Kiyoyuki Ogata, Yuto Mochimaru, Kazuma Sei.

**Project administration:** Yumi Yamamoto.

**Resources:** Mika Ogata, Yumi Yamamoto.

**Software:** Kazuma Sei, Naoya Kawahara.

**Writing – original draft:** Kiyoyuki Ogata, Yumi Yamamoto.

**Writing – review & editing:** Kiyoyuki Ogata.

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
