## [Decision Letter · Decision Letter 0]

4 Jul 2023

PONE-D-23-10670Myeloblasts transition to megakaryoblastic immunophenotypes over time in some patients with myelodysplastic syndromesPLOS ONE

Dear Dr. Ogata,

Thank you for submitting your manuscript to PLOS ONE. The paper is interesting and is of value to the field. After careful consideration, we feel that it has merit but does not fully meet PLOS ONE’s publication criteria as it currently stands. Therefore, we invite you to submit a revised version of the manuscript that addresses the points raised during the review process. Apologies for delay - we had difficulty obtaining timely responses from reviewers at this time.

Please try and address to the suggestions by both reviewers. 

We look forward to receiving your revised manuscript.

Kind regards,

Daniel Thomas, MD

Academic Editor

PLOS ONE

Additional Editor Comments:

Ogata et al present real world flow cytometry data showing a reasonable proportion of high risk MDS patients express CD41 and can acquire CD41 over time, not due to platelet adherance/micromegakaryocytes. The findings are relevant and interesting given new data indicating megakaryocyte progenitors can arise directly from HSCs. A deficiency of the manuscript is the flow cytometry data is not linked closely with clinical data.

Minor suggestions:

1] What proportion of CD41+ vs CD41- negative patients were actually t-MN, given the complex karyotypes noted. Should be discussed if no data.

2] Can we do a correlation plot of CD41 vs CD61 in the few patients tested and CD41 vs CD42b? Can we draw conclusions on what added benefit CD61 vs CD42b would have in the flow diagnostic panel?

3] Was CD110, TpoR, also expressed in any of the CD41+ blasts?

4] It would be desirable to present CD41+ (at any time) vs CD41- patients in a side by side table with P-values of significance - easy for the reader to see the differences

5] Early platelet recovery is a sign of azacitidine response - can we comment on the duration of azacitidine (+/- ven) for any of the CD41+ vs CD41- negative patients?

Reviewers' comments:

Reviewer's Responses to Questions

**Comments to the Author**

1. Is the manuscript technically sound, and do the data support the conclusions?

Reviewer #1: Partly

2. Has the statistical analysis been performed appropriately and rigorously? 

Reviewer #1: N/A

3. Have the authors made all data underlying the findings in their manuscript fully available?

Reviewer #1: Yes

4. Is the manuscript presented in an intelligible fashion and written in standard English?

Reviewer #1: Yes

5. Review Comments to the Author

Reviewer #1: Ogata and colleagues describe the expression of CD41 on CD34+ precursor cells in MDS or oligoblastic AML. This is not new, but extends on previous work from the same group. The main novelty is the report that acquisition of CD41 expression during azacytidine therapy (n=7) was associated with poor response or progression on therapy.

Major comment

The way that this study was undertaken introduces substantial bias. The patients were eligible for HMA therapy, so enriched for high risk. Serial bone marrow aspirates were performed only for poor responders and not systematically for all patients. This means that we simply don’t know whether CD41 expression might also arise in lower risk patients or those apparently responding to HMA. This should be acknowledged as a confounding factor.

Minor comment

Some of the references are incomplete (online prepublication details only)

6. PLOS authors have the option to publish the peer review history of their article (what does this mean?). If published, this will include your full peer review and any attached files.

Reviewer #1: No

---

## [Author Response · Author response to Decision Letter 0]

11 Aug 2023

Dear Dr. Thomas,

Thank you for reviewing and commenting on our manuscript titled “Myeloblasts transition to megakaryoblastic immunophenotypes over time in some patients with myelodysplastic syndromes.” We appreciate your valuable time and efforts.

Here, I am resubmitting our manuscript, modified according to the comments from the editor and reviewer as well as the style requirements of PLOS ONE. The modified parts are highlighted in red in the revised manuscript.

The original comments from the editor and reviewer and our responses to them are presented below in a point-by-point manner. 

We hope that the revised manuscript is now acceptable for publication in PLOS ONE. 

Thank you for your consideration. I look forward to hearing from you.

Kiyoyuki Ogata, MD

 

Responses to the editor

The editor’s comment 1

What proportion of CD41+ vs CD41- negative patients were actually t-MN, given the complex karyotypes noted. Should be discussed if no data.

Our response

Ten patients in our cohort had therapy-related myeloid neoplasms (t-MN), which was unrelated to their CD41-positivity. In this revision, we showed these data in Table 1 and S2 Table. 

The editor’s comment 2

Can we do a correlation plot of CD41 vs CD61 in the few patients tested and CD41 vs CD42b? Can we draw conclusions on what added benefit CD61 vs CD42b would have in the flow diagnostic panel?

Our response

We examined the expression of CD41, CD42b, and CD61 on blasts in the same tube using flow cytometry; this was respectively done for five CD41+ patients, as described in the Methods and Table 2. In all five patients, the expression level of CD41 and CD61 on blasts showed a very strong correlation as shown in Figure 3 (Case 6). In the revised manuscript, we have added the plot of CD41 vs. CD61 for the other four cases (S2 Fig). Such a strong correlation between CD41 and CD61 expression is also observed in normal platelets (Ann Lab Med. 2014. 34(6): 471–474), as CD41 and CD61 molecules are known to form a gpIIb/IIIa complex.

Further, in our five cases, addition of the CD61 vs. CD42b plot had no benefit in diagnosing patients with CD41+ MDS. However, because the number of cases was small, we avoided drawing a conclusion regarding the diagnostic value of adding CD61 (or the CD61 vs. CD42b plot); instead, we showed the CD41 vs. CD61 plots from all five cases as described above and added a comment on this issue (pages 14 and 18).

As different laboratories are using different antibody panels in diagnostic flow cytometry, the data of CD61 expression in addition to those of CD41 and CD42b shown in this manuscript can be helpful to some laboratories.

The editor’s comment 3

Was CD110, TpoR, also expressed in any of the CD41+ blasts?

Our response

This is an important issue from various viewpoints, including the positive or negative effect of TpoR agonists on patients with CD41+ MDS. Unfortunately, we have not yet examined TpoR expression in any of the cases.

The editor’s comment 4

It would be desirable to present CD41+ (at any time) vs CD41- patients in a side by side table with P-values of significance - easy for the reader to see the differences.

Our response

In this revision, we created a new table (S2 Table), wherein basic characteristics were compared between CD41+ and CD41- patients in a side-by-side fashion with P-values. 

The editor’s comment 5

Early platelet recovery is a sign of azacitidine response - can we comment on the duration of azacitidine (+/- ven) for any of the CD41+ vs CD41- negative patients?

Our response

In this revision, we compared the number of azacitidine (+/- ven) cycles administered between the CD41+ and CD41- patients. More specifically, the number of cycles until the detection of CD41-positivity in former patients and that until the last flow cytometry analysis in latter patients were compared. However, we did not find a significant difference in the number (S2 Table).

Responses to the reviewer

Thank you very much for reviewing and commenting on our manuscript, which helped improve our manuscript. We highly appreciate your valuable time.

The reviewer’s major comment

The way that this study was undertaken introduces substantial bias. The patients were eligible for HMA therapy, so enriched for high risk. Serial bone marrow aspirates were performed only for poor responders and not systematically for all patients. This means that we simply don’t know whether CD41 expression might also arise in lower risk patients or those apparently responding to HMA. This should be acknowledged as a confounding factor.

Our response

We completely agree with this comment. We have acknowledged this important point in the discussion of the revised manuscript (page 19-20).

The reviewer’s minor comment

Some of the references are incomplete (online prepublication details only).

Our response

We have examined and corrected the references thoroughly.

---

## [Editor Report · Decision Letter 1]

4 Sep 2023

Myeloblasts transition to megakaryoblastic immunophenotypes over time in some patients with myelodysplastic syndromes

PONE-D-23-10670R1

Dear Dr. Ogata,

We’re pleased to inform you that your manuscript has been judged scientifically suitable for publication and will be formally accepted for publication once it meets all outstanding technical requirements.

Kind regards,

Daniel Thomas, MD

Academic Editor

PLOS ONE

Additional Editor Comments (optional):

All comments have been satisfactorily addressed
---

## [Editor Report · Acceptance letter]

8 Sep 2023

PONE-D-23-10670R1 

Myeloblasts transition to megakaryoblastic immunophenotypes over time in some patients with myelodysplastic syndromes 

Dear Dr. Ogata:

I'm pleased to inform you that your manuscript has been deemed suitable for publication in PLOS ONE. Congratulations! Your manuscript is now with our production department. 

Kind regards, 

on behalf of

Dr. Daniel Thomas 

Academic Editor

PLOS ONE